# Heart Rehabilitation in patients awaiting Open heart surgery targeting to prevent Complications and to improve Quality of life (Heart-ROCQ): study protocol for a prospective, randomised, open, blinded endpoint (PROBE) trial

Johanneke Hartog,[1] Fredrike Blokzijl,[1] Sandra Dijkstra,[1] Mike J L DeJongste,[2] Michiel F Reneman,[3] Willem Dieperink,[4] Iwan C C van der Horst,[4] Joke Fleer,[5] Lucas H V van der Woude,[3,6] Pim van der Harst,[7] Massimo A Mariani[1]

For numbered affiliations see end of article.

**Correspondence to**
Johanneke Hartog;
j.hartog@umcg.nl

## ABSTRACT

**Introduction** The rising prevalence of modifiable risk factors (eg, obesity, hypertension and physical inactivity) is causing an increase in possible avoidable complications in patients undergoing cardiac surgery. This study aims to assess whether a combined preoperative and postoperative multidisciplinary cardiac rehabilitation (CR) programme (Heart-ROCQ programme) can improve functional status and reduce surgical complications, readmissions and major adverse cardiac events (MACE) as compared with standard care.

**Methods and analysis** Patients (n=350) are randomised to the Heart-ROCQ programme or standard care. The Heart-ROCQ programme consists of a preoperative optimisation phase while waiting for surgery (three times per week, minimum of 3 weeks), a postoperative inpatient phase (3 weeks) and an outpatient CR phase (two times per week, 4 weeks). Patients receive multidisciplinary treatment (eg, physical therapy, dietary advice, psychological sessions and smoking cessation). Standard care consists of 6 weeks of postsurgery outpatient CR with education and physical therapy (two times per week). The primary outcome is a composite weighted score of functional status, surgical complications, readmissions and MACE, and is evaluated by a blinded endpoint committee. The secondary outcomes are length of stay, physical and psychological functioning, lifestyle risk factors, and work participation. Finally, an economic evaluation is performed. Data are collected at six time points: at baseline (start of the waiting period), the day before surgery, at discharge from the hospital, and at 3, 7 and 12 months postsurgery.

**Ethics and dissemination** This study will be conducted according to the principles of the Declaration of Helsinki (V.8, October 2013). The protocol has been approved by the Medical Ethical Review Board of the UMCG (no 2016/464). Results of this study will be submitted to a peer-reviewed scientific journal and can be presented at national and international conferences.

## Strengths and limitations of this study

► This is a prospective, randomised, open, blinded endpoint trial.
► This study is a comparison of a unique multidisciplinary preoperative and postoperative cardiac rehabilitation (CR) programme with a standard care postoperative outpatient CR programme.
► The study is patient-oriented; primary and secondary outcomes provide a comprehensive and clinically relevant evaluation of patients' recovery from surgery (eg, on physical, psychological and social domains).
► Both short-term and long-term effects of the CR programmes are included.
► This is a single-centre study, although patients are referred from four hospitals.

**Trial registration number** NCT02984449.

## INTRODUCTION

The leading cause of death in Western countries is ischaemic heart disease.[1] One of the treatments for severe ischaemic heart disease is cardiac surgery. The risk of postoperative complications related to cardiac surgery is substantial; pulmonary complications (up to 33%), delirium (~26%) and arrhythmias (~30%) have been reported to occur.[2–4] These complications are associated with prolonged hospitalisation, increased adverse events (ie, readmission, stroke, myocardial infarction and mortality), reduced health-related quality of

life (HRQoL) and higher healthcare costs.[5–10] Patients with poor dietary habits (present in ~80% of the candidates for cardiac surgery), who are physically inactive (~45%), who smoke (22%), or who experience depression and/or anxiety disorders (~30%) are at higher risk for postoperative complications and are at risk for lack of functional benefits after cardiac surgery.[8 11–19]

Over the last decades risk factors such as age, obesity, diabetes, hypertension and dyslipidaemia have steadily increased in patients undergoing cardiac surgery.[20 21] Despite their adverse effects on treatment outcomes, reducing the burden of modifiable risk factors is currently not part of standard clinical care before and after cardiac surgery. Before cardiac surgery, patients often have a preoperative period of several weeks on the waiting list in which they receive little or no guidance. This waiting period has been associated with increased psychological stress, feelings of anxiety and reduced functional status.[22 23] With respect to inactivity during hospitalisation after cardiac surgery, research has shown that during the 8–11 days of hospitalisation, patients spent the majority of their time sitting or in a supine position.[24 25] In-hospital physical inactivity is a predictor of a longer hospital stay and rehospitalisation.[24 26 27] It causes a decrease in muscle strength and aerobic capacity, both fundamental in the performance of activities of daily living.[28 29] This reduced physical capacity may seriously impact independence, especially since these patients are often elderly and the functioning of their entire physiological system is already reduced.[30]

The goal of cardiac rehabilitation (CR) is to improve the preoperative and postoperative status of patients undergoing cardiac surgery. CR aims 'to favourably influence the underlying cause of cardiovascular disease, as well as to provide the best possible physical, mental, and social conditions'.[31] Postoperative CR is already an essential part of standard care in the Netherlands, although many hospitals in the Netherlands provide a phase II outpatient CR programme starting 3–6 weeks after cardiac surgery.[32 33] Benefits of postoperative CR are reported for a variety of cardiac patients[34]; however, the evidence in patients undergoing cardiac surgery is still lacking with regard to patient-relevant outcomes and mortality.[35] In addition to postoperative CR, small trials suggested that preoperative CR is effective in reducing postoperative pulmonary complications, duration of hospital stay, improving HRQoL and physical fitness, and increasing the compliance to postoperative CR.[23 36 37] However, long-term effects and the effects on other complications remain unclear. Furthermore, most studies investigated the effect of preoperative CR *or* postoperative CR, but not the effect of both CR methods combined. The hypothesis is that a combined preoperative and postoperative CR programme is more beneficial when compared with a separate preoperative CR programme *or* a single postoperative CR programme.

The aim of the Heart-ROCQ (Heart Rehabilitation in patients awaiting Open heart surgery targeting to prevent Complications and to improve Quality of life) study is to determine the effect of a comprehensive preoperative and postoperative CR programme (the Heart-ROCQ programme) on functional status, postoperative surgical complications, readmissions to hospital and major adverse cardiac events (MACE) compared with a regular Dutch postoperative outpatient CR programme (the standard care CR programme).[32] In addition, the effect of the Heart-ROCQ programme on physical and psychological functioning, including HRQoL, lifestyle risk factors, work participation and cost-effectiveness, is evaluated in comparison with the standard care CR programme. To assess who will benefit from CR and why the CR programme is effective, moderator and mediator analyses are performed.

## METHODS AND ANALYSIS
### Study design and organisation
This investigator-initiated, prospective, randomised, open, blinded endpoint trial is executed in a single centre (University Medical Center Groningen (UMCG)). Patients from three referral hospitals (Ommelander Hospital Groningen, Martini Hospital Groningen and Wilhelmina Hospital Assen) who are accepted for cardiac surgery at the UMCG are also approached to participate. Patients are randomly assigned to a combined preoperative and postoperative multidisciplinary CR programme (the Heart-ROCQ group) or a regular phase II outpatient CR programme after surgery (the Standard Care group).[32] Figure 1 provides an overview of the study design.

### Ethical considerations and dissemination
The study has been registered at ClinicalTrials.gov. If applicable, substantial amendments will be notified to the Medical Ethical Review Board (METc) for approval, and changes will be written into articles describing the results of the study. Results of this study will be submitted to a peer-reviewed scientific journal and can be presented at national and international conferences.

### Participants
Patients (≥18 years) admitted for elective coronary artery bypass grafting (CABG), valve surgery, aortic surgery or combined procedures are eligible. Patients accepted for congenital procedures, transcatheter aortic valve implantations, aortic dissections or aortic descending repair are excluded. Other exclusion criteria are inability to participate in all programme elements of the Heart-ROCQ programme due to disorders of the nervous or musculoskeletal system that limit exercise capacity, chronic obstructive pulmonary disease with Global Initiative for Chronic Obstructive Lung Disease criteria classification 3 or 4,[38] addiction to alcohol or drugs, a serious psychiatric illness (ie, recently experienced psychosis, bipolar disorder, diagnosis of schizophrenia, serious cognitive or neurological problems, and acute suicidal ideations or behaviour),

or when it is undesirable to exercise (ie, hypertrophic cardiomyopathy, unstable angina, advice from cardiologist); any treatment which is planned during one of the CR programmes and which is expected to interrupt attendance to the CR programme (eg, planned organ transplantation, preoperative endocarditis, planned chemotherapy and so on); playing a sport at the (inter) national level; and being unable to read, write or understand Dutch.

### Study enrolment, randomisation and registry
Figure 2 shows the flow chart of the Heart-ROCQ study. Patients on the waiting list of the thoracic surgery department and meeting the study criteria for type of surgery are asked by their cardiologist to participate. The cardiologist provides the patients with study information and an invitation to meet the researcher at the preoperative consultation. At the preoperative consultation, the researcher will obtain informed consent and conduct the baseline measurements. Eligible patients who have signed informed consent and performed the baseline measurements are randomised to the Heart-ROCQ group or Standard Care group. Randomisation (concealed group allocation in REDCap, random blocks of 2–4, 1:1 ratio) is stratified for weight of the surgery (isolated CABG, single non-CABG (ie, replacement or repair of part of aortic or valve), two procedures or three procedures), gender and age (≥65 and <65 years). Prior to the start of the study, the randomisation lists were created (using the 'ralloc' function of Stata/SE V.13.0) and imported into the secure data collection tool REDCap (V.7.3.2) by an independent researcher. Medical staff and researchers are not blinded to group allocation due to logistic reasons. The primary endpoint is evaluated by an independent endpoint committee blinded to group allocation.

Patients who are not willing to participate are asked to give written consent for using data that are collected during routine care. The data are collected in the Heart-ROCQ study registry to increase insight into potential differences between patients who participated in the study and patients who did not. The Heart-ROCQ registry will thus provide more insights into the generalisability of the results. Data from this registry are not used for the primary statistical analyses.

### Intervention
#### Heart-ROCQ group
The Heart-ROCQ group receives a CR programme at the Centre for Rehabilitation of the UMCG (Beatrixoord location, which is located 6 km from the hospital of the UMCG) consisting of three phases. The first phase is an outpatient preoperative optimisation phase during the waiting period (three times per week, minimum of 3 weeks). The second phase is a postoperative inpatient CR phase (3 weeks, weekdays only), followed by the third phase, an outpatient CR phase (two times per week, for 4 weeks). During each phase, all participants receive physical therapy including group sessions of inspiratory muscle training,[39] strength training, aerobic cycling, breathing, coughing and relaxation. In addition, patients have an assessment with a dietitian and a psychologist and take part, when indicated, in individual sessions to optimise their health. In addition, different group education sessions are organised regarding coping with stress, awareness of risk factors and maintaining a healthy lifestyle. Two additional modules, namely coaching to stop smoking and to return to work, are available for patients who, respectively, smoke or are employed. A detailed description of the CR programme is given in the online supplementary information.

#### Standard Care group
In the Netherlands, the current standard of care consists of a phase II outpatient CR programme which is conducted at the referred hospital.[32 33] In general, this CR programme starts 3–6 weeks after discharge from the hospital and lasts for 6 weeks. The programme consists of four educational sessions (regarding risk factors and retaining a healthy lifestyle) and physical therapy (two times per week: 30 min cycling, together with 30 min sports and games, relaxation therapy, or strength training). Furthermore, patients can be

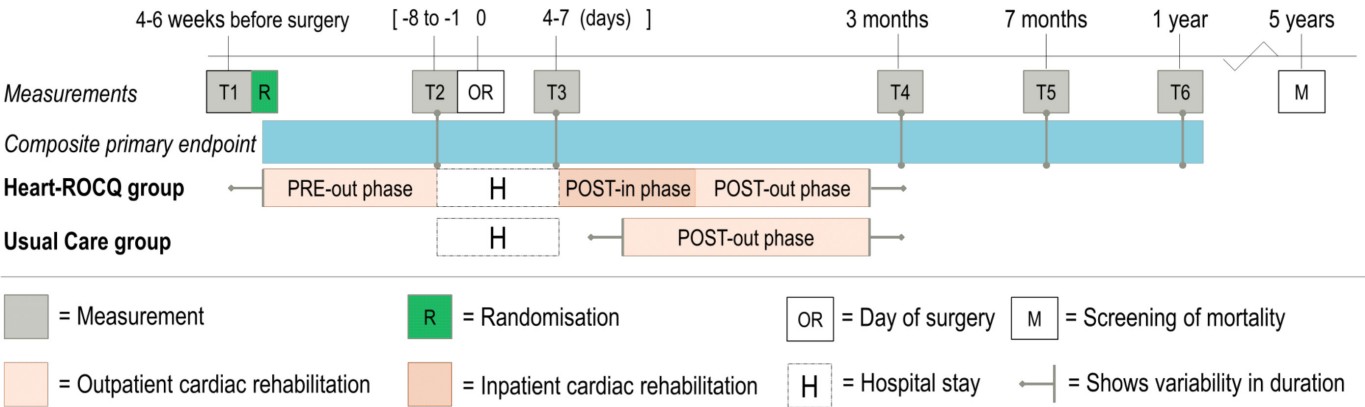

**Figure 1** Research design of the Heart-ROCQ prospective, randomised, open, blinded endpoint trial. The phases of both cardiac rehabilitation programmes and the measurements are shown relative to the moment of surgery. Heart-ROCQ, Heart Rehabilitation in patients awaiting Open heart surgery targeting to prevent Complications and to improve Quality of life.

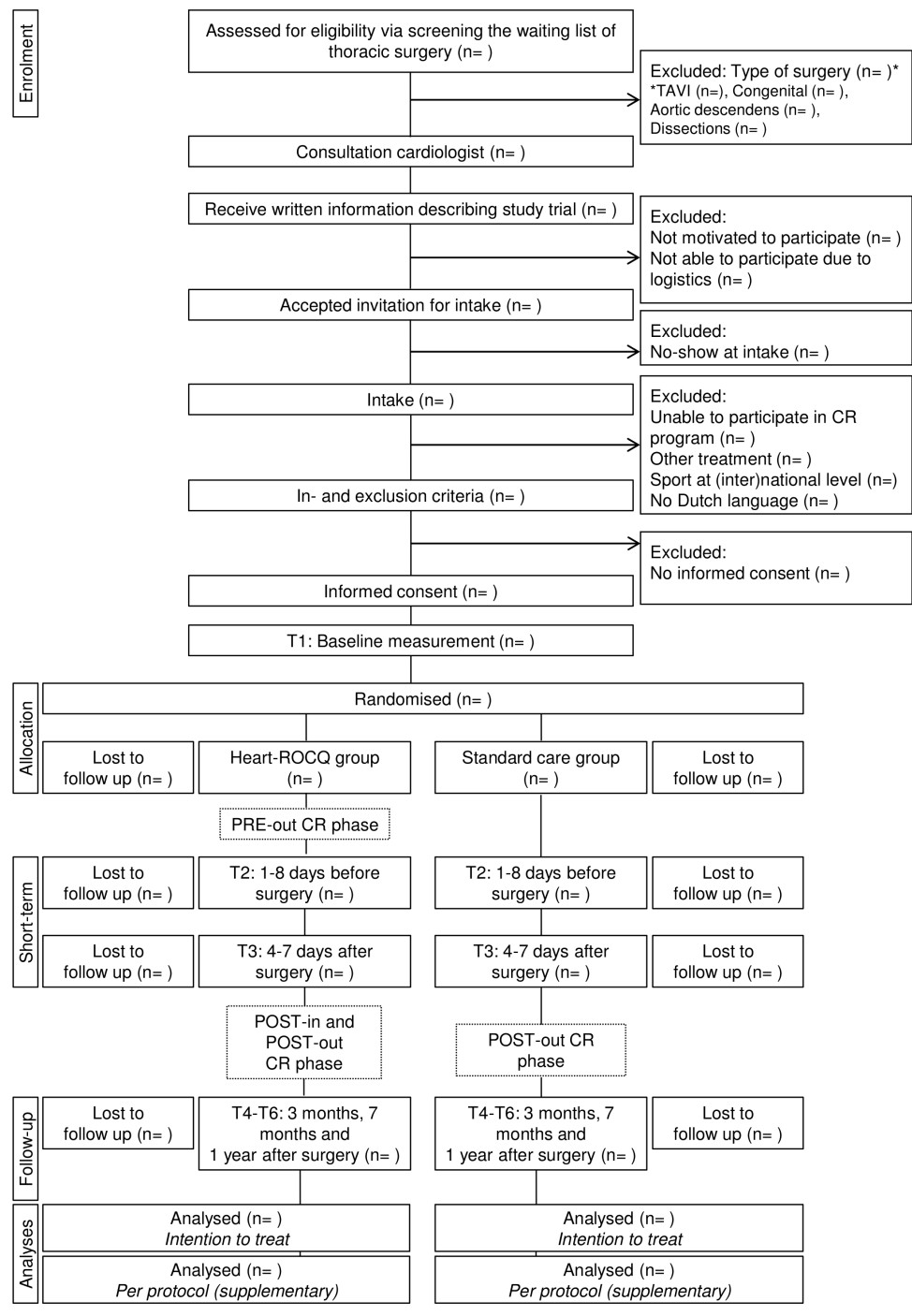

**Figure 2** Flow chart of the Heart-ROCQ study. CR, cardiac rehabilitation; Heart-ROCQ, Heart Rehabilitation in patients awaiting Open heart surgery targeting to prevent Complications and to improve Quality of life; TAVI, transcatheter aortic valve implantation.

referred to a psychologist or a dietitian when needed. The content of this CR programme is based on the Dutch CR guidelines.[40]

## Composite primary endpoint: functional status, complications and events

The primary outcome is a composite weighted score of functional status, postoperative surgical complications, readmissions to the hospital and MACE. Each event is scored (ranging from 1 to 3 points) separately. Table 1

shows an overview of the primary outcome and the scoring system.

The scores of all events are combined to calculate a total score. Only the most serious complication is counted per event (eg, when a percutaneous coronary intervention (score 1) is complicated by a stroke (score 2), the score will be 2 and not 3 (1+2)).

The concept of the composite weighted score is adapted from the African-American Heart Failure Trial.[41]

**Table 1** Definitions and score of the components of the composite primary endpoint

| Functional status | Score |
|---|---|
| Worsening in physical functioning (domain score of health-related quality of life, RAND-36 V.2).* | 1 |
| No change or improvement in physical functioning.* | 0 |
| Worsening in physical problem (domain score of health-related quality of life, RAND-36 V.2).* | 1 |
| No change or improvement in physical problem.* | 0 |
| A clinically relevant worsening is classified as minimal change according to Wyrwich et al.[59] | |
| **(Serious) adverse events** | |
| No serious adverse event. | 0 |
| Prolonged mechanical ventilation | 1 |
| Mechanical ventilation longer than 24 hours. | |
| Lung infection | 1 |
| (1) A new lung radiographic infiltrate and (2) two signs that the infiltrate is of infectious origin—that is, (a) fever: body temperature >38°C or <36°C; (b) leucocytosis—white cell counts >$10 \times 10^9$/L or <$4 \times 10^9$/L; (c) positive sputum culture; and/or (d) decline in oxygenation. | |
| Delirium | 1 |
| (1) A DOS[60] score ≥3 at hospital ward and/or (2) diagnosis confirmed by a psychiatrist, geriatrist or supervising specialist according to the DSM-IV criteria resulting in treatment with medication. | |
| Readmissions to intensive care unit | 1 |
| Unrelated to a secondary endpoint. | |
| Deep wound infection | 2 |
| Deeper tissues are affected (muscle, sternum and mediastinum) and must include (1) surgical drainage/refixation or (2) an organism is isolated from culture of mediastina tissue or fluid, or (3) antibiotic treatment because of sternum wound. | |
| Readmissions to hospital | 1 |
| An unplanned hospital stay with different dates of admission and discharge with a medical indication (ie, clinical signs and symptoms or change of treatment). | |
| Any cardiothoracic surgical interventions | 2 |
| Graft or valve failure, CABG, valve, aortic or other cardiac surgery.† | |
| Any percutaneous interventions | 1 |
| PCI, TAVI and so on. | |
| Myocardial infarction | 2 |
| According to the third universal definition of myocardial infarction.‡ | |
| Cerebral vascular accident / stroke | 2 |
| Acute neurological event of at least 24 hours in duration, with focal signs and symptoms and without evidence supporting any alternative explanation. Diagnosis of stroke requires confirmation by CT, MRI or pathological confirmation. | |
| Sudden death survivor | 2 |
| The sudden onset of symptoms, such as chest pain and cardiac arrhythmias, and ventricular tachycardia, which lead to the loss of consciousness and cardiac arrest followed by reanimation and does not lead to biological death. | |
| Death | 3 |
| All-cause mortality. | |
| Total score = sum of physical status and SAEs at 3 months and 1 year (worst score of each event). | |

In violet, postoperative complications can be scored once.
*Compared with baseline.
†According to the definitions of the 'Begeleidingscommissie Hartinterventies Nederland'.
‡Thygesen et al.[61]
CABG, coronary arterial bypass grafting;DOS, Delirium Observation Screening scale;DSM, Diagnostic and statistical manual of mental disorders; PCI, percutaneous coronary intervention;RAND-36 V.2, Medical Outcome Study 36-item General Health Survey; SAE, serious adverse event; TAVI, transcatheter aortic valve implantation.

Functional status is assessed through two health domains of the Medical Outcomes Study 36-item General Health Survey (RAND-36 V.2)[42]: physical functioning and physical health problems. The primary endpoint is evaluated at 3 months (T4) and 1 year (T6) after surgery (figure 1). Deep wound infections and surgical re-explorations are screened up to 120 days after surgery. Other postoperative surgical complications are measured in the period between the surgery and when a patient meets the UMCG discharge criteria (box 1). Hospital admissions are checked between baseline and 1 year after surgery. To prevent bias (since the Heart-ROCQ group follows the

<table>
<tr><td>Box 1</td><td>Discharge criteria of the University Medical Center Groningen</td></tr>
</table>

► No drain, no external pacemaker lead, no infusion or oxygen present.
► Stable clinical conditions (stable lab results, X-ray and haemodynamic parameters).
► Able to perform basic activity of daily living activities (ie, going independently to the toilet).

inpatient CR phase after surgery), hospital admissions between the day of admission before surgery and 30 days after surgery are not included when determining the (calculated) primary endpoint.

## Secondary outcomes
### Complications and events

All individual components of the composite endpoint regarding the complications and events will be analysed separately as secondary outcomes. Table 2 summarises the definitions of the secondary complications and events, including at the time of screening.

All documents concerning the composite primary endpoint and the secondary complications are encrypted and subsequently adjudicated by the independent endpoint committee. The endpoint committee consists of four members (cardiologists and cardiothoracic surgeons) who are not employed in the UMCG and are blinded to group allocation.

**Table 2** Definitions of the secondary complications and events

| Definitions | Time of measure |
|---|---|
| Atrial fibrillation | Surgery to T3 |
| New onset of atrial fibrillation or atrial flutter requiring medical treatment or cardioversion. | |
| Prolonged ICU stay | Initial stay |
| When the number of calendar days is two or more from ICU admission to discharge. | |
| Readmissions to hospital | Baseline to T6* |
| The number of unplanned hospital stays with different dates of admission and discharge with a medical indication (ie, clinical signs and symptoms or change of treatment). a. Directly related to a cardiac cause. b. Not directly related to a cardiac cause. | |
| Hospitalisation days | Baseline to T6* |
| Total number of days of hospitalisation. a. Directly related to a cardiac cause. b. Not directly related to a cardiac cause. | |
| Cardiovascular death | Baseline to 5 years after surgery |
| Any death due to proximate cardiac or cardiovascular cause (eg, myocardial infarct, low-output failure, fatal arrhythmia, death secondary to a cerebral vascular accident, pulmonary embolism, ruptured or dissecting aortic aneurysm, or other vascular diseases), unwitnessed death, and death of unknown cause, and all procedure-related deaths (eg, PCI, CABG), including those related to concomitant treatment†. | |
| Non-cardiovascular death | Baseline to 5 years after surgery |
| Any death not covered by the above definition, such as death caused by infection, malignancy, sepsis, pulmonary causes, accident, suicide or trauma. | |
| **Concerning safety** | |
| Surgical re-exploration for bleeding/ tamponade | Surgery to T4 |
| Surgical incision into the sternum as a result of a bleeding or tamponade. a. Acute: presented within 24 hours after surgery. b. Late: presented after 24 hours after surgery. | |
| Surgical re-exploration dehiscence | Surgery to T4 |
| Aseptic wound dehiscence. | |

*Not measured between the day of admission before surgery and 30 days after surgery because of POST-in phase of the Heart-ROCQ programme.
†Specifically, any unexpected death even in patients with coexisting, potentially fatal non-cardiac disease (eg, cancer, infection) will be classified as cardiac death.
CABG, coronary artery bypass grafting; Heart-ROCQ, Heart Rehabilitation in patients awaiting Open heart surgery targeting to prevent Complications and to improve Quality of life; ICU, intensive care unit; PCI, percutaneous coronary intervention; T4, follow-up at 3 months after surgery; T6, follow-up at 12 months after surgery.

**Table 3** Time of measure, physical tests and questionnaires of the secondary outcomes

| Secondary outcomes | Measure | Time of measure |
|---|---|---|
| **Physical health** | | |
| Cardiorespiratory fitness | 6MWT[62] | T1–T4, T6 |
| Muscle strength | STS-10, grip and leg strength[63 64] | T1–T4, T6 |
| Independence in ADL | KATZ[65 66] | T1, T4, T6 |
| **Psychological health** | | |
| General anxiety | GAD-7[67] | T1, T2, T4, T6 |
| Feelings of depression | PHQ-9[68 69] | T1, T2, T4, T6 |
| Health-related quality of life | RAND-36 V.2[42] | T1, T4, T6 |
| **Lifestyle risk factors** | | |
| Physical activity | iPAQ[70] and Actigraph*[71] | T1, T4, T6 |
| Obesity indices | BMI, waist to hip ratio | T1, T4, T6 |
| Smoking behaviour | Number of cigarettes per day | T1, T4, T6 |
| **Economic evaluation** | | |
| Healthcare use and related medical costs | iMCQ[72] | T1, T4–T6 |
| Work participation† | iPCQ[73] | T1, T4–T6 |
| QALYs | EQ-5D-5L[52] | T1, T4–T6 |
| **Potential mediators** | | |
| Cardiac self-efficacy | CSE[45] | T1, T2, T4, T6 |
| Illness representations | IPQ-R[43] | T1, T2, T4, T6 |

*Patients are asked to wear the Actigraph during waking hours for one consecutive week.
†Health-related productivity losses of paid work and unpaid work.
ADL, activities of daily living; BMI, body mass index; CSE, cardiac self-efficacy; EQ-5D-5L, EuroQol Five-Dimensional Questionnaire; GAD-7, Generalised Anxiety Disorder 7-item scale; iMCQ, iMTA Medical Cost Questionnaire; iPAQ, International Physical Activity Questionnaire; iPCQ, iMTA Productivity Cost Questionnaire; IPQ-R, Illness Perception Questionnaire, Revised; KATZ, Katz Index of Independence in Activities of Daily Living; 6MWT, 6 min walking test; PHQ-9, Patient Health Questionnaire 9-item scale; QALY, quality-adjusted life years; RAND-36 V.2, Medical Outcome Study 36-item General Health Survey; STS-10, 10 times sit to stand test; T1, begin of waiting time; T2, 1–8 days before surgery; T3, moment that patients meet the University Medical Center Groningen discharge criteria; T4–T6, follow-up at 3, 7 and 12 months after surgery.

### Questionnaires and physical tests

Table 3 shows the physical tests and questionnaires of the secondary outcomes regarding physical and psychological health, lifestyle risk factors, and the economic evaluation. Physical tests and questionnaires are completed at six assessment points (figure 1). The preoperative measurements (T1 and T2) are conducted at the start (preoperative consultation) and the end (1–8 days before surgery) of the waiting period. The third measurement (T3) is performed when patients meet the UMCG discharge criteria (box 1). The follow-up measurements (T4, T5 and T6) are at 3, 7 and 12 months after surgery. Patients are asked to fill in questionnaires online, or when requested on paper prior to the visits for the physical tests. All adverse events reported spontaneously by the patient or observed by the investigator are recorded. In addition, serious adverse events are reported to the METc. Data are stored in REDCap and on a secure drive at the CardioResearch department in the UMCG. Two times per year, the study is monitored by a trained research monitor from another department of the UMCG. Details of procedures, data collection, management and monitoring can be found in the Trial Master File of the Heart-ROCQ study (can be obtained by the investigators). On 12 June 2018, the study was audited and certified for ISO 9001:2015 independently by DNV GL.

### Potential moderators

Preoperative risk profile (ie, Euroscore II, medical history, lifestyle risk factors, and physical and mental status), surgery parameters and demographics are collected from the medical record and baseline measurements. The content of the CR programme is described in terms of compliance, duration of CR programme, type of treatment, frequency and training load (ie, for bicycle training: external workload, heart rate response, and rate of perceived exertion using the Borg Scale; and for strength training: sets, repetitions and intensity) for both the Heart-ROCQ group and the Standard Care group.

### Potential mediators

Illness perceptions (ie, patients' mental representations of their illness based on different sources of information) are measured with three subscales (personal control,

treatment control and consequences) of the Revised Illness Perception Questionnaire.[43] These subscales are chosen because of their sensitivity to change and because of their relation to psychological distress.[44] The Cardiac Self-Efficacy (CSE) Scale[45] is used to measure CSE (ie, patients' confidence to perform a specific task related to cardiac disease).

## Statistical analyses

### Sample size

Assuming a normal distribution, the mean weighted score of the primary endpoint is estimated at 1.0 with an SD of 0.9 at 1 year after surgery. This estimation is based on historical data of the UMCG, an unpublished ongoing pilot study in the UMCG, the Dutch registration database[46] and data reported in the literature.[6 47–49] A decrease of 0.3 is expected in the Heart-ROCQ group, based on previous studies comparing CR with standard care[36 48 50] (ie, no CR), and is considered to be clinically relevant (ie, on average a 30% decrease in complications/events or worsening in functional status, or 10%–20% decrease in MACE or death). To detect this decrease and achieve 80% power (significance level of 5%), a group of 286 patients (143 in the Heart-ROCQ group and 143 in the Standard Care group) is needed. To incorporate a withdrawal of ±20%, a total sample size of 350 is needed at baseline.

### Interim analysis

An interim analysis will be conducted when 40% of the included patients have had the measurements 1 year after surgery. The study will be terminated prematurely when the primary outcome of one of the CR programmes is obviously (p<0.001) different from the other CR programme.

### Primary and sensitivity analyses

All endpoints are primarily analysed according to the 'intention-to-treat' principles, and missing values are counted as worst-case score (nominal variables) or estimated using maximum likelihood estimation (interval variables). As supplementary analyses, the endpoints are analysed on a per-protocol principle with and without using imputation methods for missing values. In all analyses, a two-sided p<0.05 is considered statistically significant.

The total score of the primary endpoint will be handled as a continuous variable. All continuous variables will be analysed using linear mixed models to determine 'time x group' differences. Significant interactions will be further explored using the Bonferroni post-hoc test to determine differences between each time point. Non-parametric tests will be used if the assumptions of normal distribution are violated. More information about the statistical methods and clinical relevance are written in the research protocol (ClinicalTrials.gov: NCT02984449).

### Economic evaluation

For the evaluation of healthcare utilisation, standard prices published in the Dutch costs guidelines are used.[51] To compare the costs with quality-adjusted life years (QALYs), QALYs are estimated with the use of the EuroQol Five-Dimensional Questionnaire (EQ-5D-5L).[52] Utility values for the EQ-5D-5L are calculated based on the new Dutch tariff.[53] Results from the analysis are reported as an incremental cost-effectiveness ratio, dividing the difference in effect by the difference in costs. Bootstrap resampling will be performed, and cost-effectiveness acceptability curves will be plotted, to estimate the probability that the Heart-ROCQ programme is cost-effective when compared with standard care. A societal perspective is applied.

## Study status

From May 2017 to December 2018, 75 patients were enrolled. In the following years we expect that the enrolment will increase to 85 patients per year. The last patient is expected to be included in July 2021.

## Patient and public involvement

In a pilot study (to be submitted), patient satisfaction and feasibility (in terms of accessibility, compliance, training load and safety) of the Heart-ROCQ programme have been evaluated. Patients were very satisfied with the programme, scoring it 8 out of 10; therefore, we did not change the content of the programme. However, patients' rate of perceived exertion was generally quite low and no serious adverse events occurred during the bicycle training. For safety reasons the intensity was not increased. However, in order to estimate the maximum load more accurately and better tailor the programme to the individual, we decided to change one of the stop criteria of the preoperative submaximal ergometry test from 70% to 90% of the expected maximal heart rate. Furthermore, our outcomes are, among others, based on the reasons why patients recommended the programme to other patients. For example, patients reported that their self-efficacy and physical capacity were improved, so we added the CSE questionnaire and physical tests to objectively measure these outcomes. In this way the results were taken into account in the further development of the Heart-ROCQ programme and the protocol of this study. The results of this trial will be distributed by various information channels (eg, websites of cardiac patient organisations, social media). Two to three times a year, we provide a newsletter about the progress, and (in the end) the results of the study are sent to patients who are interested.

## DISCUSSION

The Heart-ROCQ study is the first randomised clinical trial evaluating the effect of a combination of preoperative and postoperative CR programme compared with a postoperative CR programme. Unlike the vast majority of CR programmes in previous studies, the current programme is multidisciplinary, targeting different aspects of surgical outcomes in patients undergoing cardiac surgery. Because different aspects are targeted, a

composite weighted score will better reflect the treatment benefits than a single outcome. Therefore, the primary endpoint is a composite endpoint of functional status, postoperative surgical complications, readmissions to the hospital and MACE. The components of the endpoint are of clinical importance to patients undergoing cardiac surgery and reflect a comprehensive representation of the recovery of the patient.

Both the ideas of the combined primary endpoint and the weighting of the individual components were derived from other studies.[41 54 55] Assigning different weights to the components was needed for more accurate comparison. Since the components are not equal in clinical importance, equal weights would lead to inaccurate statistical analysis.[55] In the current study, the weighted score of HRQoL is lower (1 point instead of 2 points) to minimise bias due to patients' knowledge of group allocation. Therefore, improvements in quality of life are not counted in the primary endpoint to prevent bias in a positive direction. This also prevents that a score in quality of life and an adverse event cancel each other out (eg, when a patient experiences an improvement in quality of life (+2 points) and has a stroke (−2 points), then the total score is 0). The scores of all-cause mortality, stroke, myocardial infarction and revascularisation are in line with the results shown by Tong and colleagues.[55] A disadvantage of composite endpoints is that the effect may be driven by complications that occur with the greatest frequency.[56] Therefore, postoperative complications which occur frequently, such as atrial fibrillation, are evaluated separately as secondary endpoint. The primary endpoint is evaluated by an endpoint committee, which is blinded to group allocation and consists of four independent cardiologists and cardiothoracic surgeons.

Previous preoperative CR studies were primarily focused on short-term effects; only one preoperative CR study and a few postoperative CR studies have determined long-term effects.[23 35 36 57] In contrast to previous preoperative CR studies, the Heart-ROCQ study sets out to investigate both short-term and long-term effects of the CR programmes.[23 35 36] Due to the trends in, among others, increasing age, obesity and physical inactivity, patients undergoing cardiac surgery are becoming more complex. The Heart-ROCQ programme aims to address these issues, which makes the programme clinically relevant for all cardiac surgery patients. Therefore, we chose to include different types of cardiac surgery. Since different moderators and mediators are assessed before, during and after CR, we can explore which factors are associated with better outcomes and which working mechanisms contribute to its effectiveness. These findings may provide a more indepth understanding of who benefits the most from CR in both the short and long term and the underlying mechanisms of CR, which are still not fully understood in patients undergoing cardiac surgery.[35] In addition, the present study is thought to considerably contribute to the evidence to further develop guidelines for clinical practice, especially regarding the preoperative CR programme.[58]

The Heart-ROCQ programme is expected to be cost-effective in the long term, which is also of interest for policymakers and healthcare providers. Therefore, an economic evaluation is performed to assess the cost-effectiveness of CR, since little is known about the cost-effectiveness of CR.[48] A societal perspective of this economic evaluation is chosen, meaning that healthcare costs and patient-related and productivity-related costs and benefits are taken into account. If the Heart-ROCQ programme is proven to be effective, it might be advisable to include a supportive prevention-oriented waiting period prior to surgery as an integral part of the treatment for patients undergoing elective cardiac surgery. This implies a paradigm shift from curative care following cardiac surgery to an additional preventive care attitude before surgery. Extensions of rehabilitation options in or in the vicinity of cardiac centres will then be required.

The Heart-ROCQ study is the first randomised clinical trial comparing the effect of a combined preoperative and postoperative CR programme with a regular Dutch phase II postsurgery outpatient rehabilitation CR programme in a population undergoing elective cardiac surgery. This study is expected to provide new understanding of the effectiveness and underlying working mechanisms of CR, and subsequently to improve value-based healthcare.

**Author affiliations**
[1]Department of Cardio-thoracic Surgery, University of Groningen, University Medical Center Groningen, Groningen, The Netherlands
[2]University of Groningen, University Medical Center Groningen, Groningen, The Netherlands
[3]Department of Rehabilitation Medicine, University of Groningen, University Medical Center Groningen, Groningen, The Netherlands
[4]Department of Critical Care, University of Groningen, University Medical Center Groningen, Groningen, The Netherlands
[5]Department of Health Psychology, University of Groningen, University Medical Center Groningen, Groningen, The Netherlands
[6]Center for Human Movement Sciences, University of Groningen, University Medical Center Groningen, Groningen, The Netherlands
[7]Department of Cardiology, University of Groningen, University Medical Center Groningen, Groningen, The Netherlands

**Acknowledgements** We thank EJ Beens, MJ Nijholt, Drs A Branderhorst, Dr B Dorhout (in memoriam), Dr R de Jong, HAM Wasser, Dr J Brügemann, Dr JB Wempe, Dr J Posma, Dr CA Geluk and Drs ML Pentinga for supporting the Heart-ROCQ study. In addition, we thank Drs TW Waterbolk, Dr RG Tieleman and Dr RPM van Roosmalen for their participation in the endpoint committee. We are grateful to all involved colleagues in the Department of Cardiology and Cardio-thoracic Surgery, and the Centre for Rehabilitation of the UMCG, Ommelander Hospital Groningen, Martini Hospital Groningen and Wilhelmina Hospital Assen for the pleasant collaboration and their help with executing the Heart-ROCQ study.

**Contributors** Conception and design of the study: JH, FB, MJLD, MFR, WD, JF, LHVvdW, PVdH and MAM. Methodology: JH, FB, SD, MJLD, MFR and PVdH. Acquisition of data: JH, FB and SD. Writing - original draft, tables and figures: JH. Writing - reviewing and editing: FB, SD, MJLD, MFR, WD, ICCvdH, JF, LHVvdW, PVdH and MAM. Supervision: MJLD, MFR, WD, JF, ICCvdH, LHVvdW, PVdH and MAM. Funding acquisition: JH, SD, JF, LHVvdW, PVdH and MAM. All authors approved the final version of the manuscript.

**Funding** The Heart-ROCQ study is financially supported by Edwards Lifesciences SA, Abbott (formerly St Jude Medical Nederland) and 'Stichting Beatrixoord Noord-Nederland'. The sponsors are not involved in the design or execution of the study.

**Competing interests** JH, SD and MAM report grants from Edwards Lifesciences SA, Abbott (formerly St Jude Medical Nederland) and 'Stichting Beatrixoord Noord-Nederland' related to this study. In addition, MAM reports consultancy from AtriCure, Getinge and LivaNova. JF, WD, LHVvdW, MFR, FB, ICCvdH, MJLD and PVdH have nothing to disclose.

**Patient consent for publication** Not required.

**Ethics approval** This study is conducted according to the principles of the Declaration of Helsinki (V.8, October 2013) and according to the research code of the UMCG. The protocol (V.2, 19 December 2016) has been approved by the Medical Ethical Review Board (METc) of the UMCG (no 2016/464).

**Provenance and peer review** Not commissioned; externally peer reviewed.

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
