## [Reviewer comments · BMJ Open]

ARTICLE DETAILS

TITLE (PROVISIONAL)	HEART Rehabilitation in patients awaiting Open-heart surgery targeting to prevent Complications and to improve Quality of life (Heart-ROCQ): Study protocol for a Prospective Randomised Open Blinded End-point (PROBE) trial
AUTHORS	Hartog, Johanneke; Blokzijl, Fredrike; Dijkstra, Sandra; DeJongste, Mike; Reneman, Michiel; Dieperink, Willem; van der Horst, Iwan; Fleeer, Joke; van der Woude, Lucas; Van der Harst, Pim; Mariani, Massimo

VERSION 1 – REVIEW

REVIEWER	Daniel T. Engelman Baystate Medical Center, Springfield, MA, USA
REVIEW RETURNED	05-Jun-2019

GENERAL COMMENTS	Up to 80% of preventable morbidity and mortality surrounding cardiac surgery takes place in the preoperative and postoperative timeframe.[1] It therefore follows that efforts to modify select risk factors prior to and following cardiac surgery, may result in significant improvements in outcomes. Similar interventions have already been proposed in recent guidelines for cardiac surgical perioperative care.[2] In this proposed study 350 patients will be randomized to a pre and postoperative multidisciplinary rehabilitation program versus standard care. The preoperative optimization phase appears appropriate at three times per week for a minimum of three weeks. The postoperative inpatient CR phase (3 weeks, only during weekdays) may be somewhat costly and burdensome as it extends inpatient care for three weeks. I would consider whether similar results could be obtained through an outpatient format. It will also be difficult to differentiate the contributions of the preoperative versus post-operative rehabilitation phases, as to which contributed to any positive outcome results at the conclusion of the study. The economic evaluation as it relates to health care utilization is an important component of this study and should be commended. Similarly, the multidisciplinary nature of this intervention protocol is perfectly suited for this complex vulnerable patient population. I believe this protocol is scientifically credible and is presented in an appropriate context; the design is ethically and procedurally sound. 1. Shannon, F.L., et al., A method to evaluate cardiac surgery mortality: phase of care mortality analysis. Ann Thorac Surg, 2012. 93(1): p. 36-43; discussion 43.
---

	2. Engelman, D.T., et al., Guidelines for Perioperative Care in Cardiac Surgery: Enhanced Recovery After Surgery Society Recommendations. JAMA Surg, 2019.
--	--

REVIEWER	Scott Kehler Dalhousie University, Canada
REVIEW RETURNED	14-Jun-2019

GENERAL COMMENTS	Thank you for the opportunity to review this interesting clinical trial protocol which will compare usual care of postoperative cardiac rehabilitation with a pre- and postoperative cardiac rehabilitation program in patients undergoing cardiac surgery. Optimizing risk factors prior to cardiac surgery with a “prehab” program such as this has the potential to improve patient outcomes (PMID: 10929164; PMID: 24459173). The authors intend to recruit over 300 participants to determine if their intervention will reduce a primary endpoint of composite endpoint of functional status, surgery complications, readmissions, and major adverse cardiac events. Overall this trial seems to be developed with careful thought and the authors provide a detailed report. Major comments:  -Can the authors justify why they are examining a composite outcome of a host of problems? While all of the outcomes are important, it seems odd to include functional status (SF-36) with harder endpoints such as surgical complications and death. A higher weight for death vs. physical function in the composite outcome is interesting... why wouldn't physical function be more highly weighted? The patient themselves may be more devastated if they end up in worse health after their surgical intervention. -This study protocol is similar to one previously published in BMJ Open (PMID: 25753362) and to a small study examining prehab (PMID: 24459173) which were not cited in the author's paper. Can the authors please comment on how their prehab intervention will build on these studies and the ones cited in their document (ref 23, 36)? Minor comments:  -It may be helpful to revise the document for English language to improve some clarity within the document. -Please clarify in the title that this is a study protocol. -The use of CS as an acronym throughout the paper does not help with reading the document. Can the authors please spell out cardiac surgery instead of using CS? -Where it says, “in a pilot study (to be submitted) the patient satisfaction and the feasibility (in terms of accessibility, compliance, training load and safety) of the Heart-ROCQ program have been evaluated. These results were taken into account when further developing the Heart-ROCQ program and the protocol of this study.” Can the authors please provide a summary (e.g., in a table, a box, or in text) how their study was tailored based on the pilot study in the protocol here? It would be useful for other readers to know this specifically in this document. -The authors should indicate how their primary endpoint will be analyzed throughout the trial.
---

VERSION 1 – AUTHOR RESPONSE

Reviewer(s)' Comments to Author:

Comments from Reviewer: 1

Reviewer Name: Daniel T. Engelman

Institution and Country: Baystate Medical Center, Springfield, MA, USA Please state any competing interests or state 'None declared': None declared

Up to 80% of preventable morbidity and mortality surrounding cardiac surgery takes place in the preoperative and postoperative timeframe.[1] It therefore follows that efforts to modify select risk factors prior to and following cardiac surgery, may result in significant improvements in outcomes. Similar interventions have already been proposed in recent guidelines for cardiac surgical perioperative care.[2] In this proposed study 350 patients will be randomized to a pre and postoperative multidisciplinary rehabilitation program versus standard care. The preoperative optimization phase appears appropriate at three times per week for a minimum of three weeks. The postoperative inpatient CR phase (3 weeks, only during weekdays) may be somewhat costly and burdensome as it extends inpatient care for three weeks. I would consider whether similar results could be obtained through an outpatient format. It will also be difficult to differentiate the contributions of the preoperative versus post-operative rehabilitation phases, as to which contributed to any positive outcome results at the conclusion of the study.

The economic evaluation as it relates to health care utilization is an important component of this study and should be commended. Similarly, the multidisciplinary nature of this intervention protocol is perfectly suited for this complex vulnerable patient population. I believe this protocol is scientifically credible and is presented in an appropriate context; the design is ethically and procedurally sound.

1. Shannon, F.L., et al., A method to evaluate cardiac surgery mortality: phase of care mortality analysis. *Ann Thorac Surg*, 2012. 93(1): p. 36-43; discussion 43.
2. Engelman, D.T., et al., Guidelines for Perioperative Care in Cardiac Surgery: Enhanced Recovery After Surgery Society Recommendations. *JAMA Surg*, 2019.

Reply: We are grateful for the insightful comments and appreciate the reviewer' effort. We agree on the importance of the treatment preoperative and postoperative timeframe and on the duration of the preoperative optimization phase. As stated in the discussion we think the Heart-ROCQ study will 'considerably contribute to the evidence to further develop guidelines for clinical practice, especially regarding the preoperative CR program.(73)' (P.21, Line 392). It will increase the evidence level of guidelines for perioperative care in cardiac surgery, such as the ERAS recommendations, which will further improve perioperative care and will enhance recovery after surgery. The mentioned sentence from the discussion has changed slightly in the manuscript: we added 'further' and the reference of the ERAS guidelines.

Concerning the economic evaluation of the postoperative inpatient CR phase, the reviewer is correct; it is indeed important to take the costs into account. The Heart-ROCQ study will indicate if and for whom the Heart-ROCQ program will be cost-effective. Inpatient rehabilitation directly after hospitalization may further speed up recovery and enhance post-operative result. It can save costs when patients are discharged earlier to the rehabilitation centre and thus the hospital stay is reduced (since a bed in the rehabilitation centre is much cheaper compared to a hospital bed). In addition, ~80% of the first readmissions are within 30 days[1]. The inpatient phase will thus prevent these readmissions, which will also save costs. Early inpatient rehabilitation after cardiac surgery has been shown to be safe, and to improve exercise capacity and quality of life [2-4]. Speeding up recovery will also save costs, because patients return earlier to work and so make less use of home care. An outpatient format might be interesting for lowering the costs, however, during the first six weeks after the surgery, patients are advised not to drive a car and therefore are dependent. This advice is in concert with clinical practice since the outpatient rehabilitation program is often started after six weeks. Thus, during this first important six weeks period after surgery, patients do not receive rehabilitation and tend to stay sedentary. An advantage of the inpatient phase is that therapy targeting early mobilization and oral intake of fluids and solids (an important aspect of the ERAS guidelines [5])

can be provided at a daily basis for a longer period. Therapists can better guide the patients, since they are more involved in the recovery process of the patients.

The main aim of this study is to differentiate the contributions of a combined pre- and postoperative program versus a post-operative rehabilitation phase. We will test the hypothesis that the combination of pre- and postoperative rehabilitation (together with the multidisciplinary components) will add an additional effect (one and one will be three instead of two). The measurements T2 (one day before surgery) en T3 (when patients meet the UMCG discharge criteria) have been added to evaluate the short-term effects of the preoperative phase.

We want to thank the reviewer for the kind words about the economic evaluation, multidisciplinary nature of the Heart-ROCQ program and the study protocol.

1. Iribarne A, Chang H, Alexander JH, Gillinov AM, Moquete E, Puskas JD, et al. Readmissions after cardiac surgery: experience of the National Institutes of Health/Canadian Institutes of Health research cardiothoracic surgical trials network. *Ann Thorac Surg.* 2014 Oct;98(4):1274–80.
2. Meurin P, Iliou MC, Ben Driss A, et al. Early exercise training after mitral valve repair: a multicentric prospective French study. *Chest.* 2005;128(3):1638-1644. doi:10.1378/chest.128.3.1638
3. Macchi C, Fattiroli F, Lova RM, et al. Early and late rehabilitation and physical training in elderly patients after cardiac surgery. *Am J Phys Med Rehabil.* 2007;86(10):826-834. doi:10.1097/PHM.0b013e318151fd86
4. Eder B, Hofmann P, von Duvillard SP, et al. Early 4-week cardiac rehabilitation exercise training in elderly patients after heart surgery. *J Cardiopulm Rehabil Prev.* 2015;30(2):85-92. doi:10.1097/HCR.0b013e3181be7e32
5. Ljungqvist O, Scott M, Fearon KC. Enhanced recovery after surgery a review. *JAMA Surg.* 2017;152(3):292–8.

Comments from Reviewer: 2

Reviewer Name: Scott Kehler

Institution and Country: Dalhousie University, Canada Please state any competing interests or state 'None declared': None declared.

General comments

Thank you for the opportunity to review this interesting clinical trial protocol which will compare usual care of postoperative cardiac rehabilitation with a pre- and postoperative cardiac rehabilitation program in patients undergoing cardiac surgery. Optimizing risk factors prior to cardiac surgery with a "prehab" program such as this has the potential to improve patient outcomes (PMID: 10929164; PMID: 24459173). The authors intend to recruit over 300 participants to determine if their intervention will reduce a primary endpoint of composite endpoint of functional status, surgery complications, readmissions, and major adverse cardiac events. Overall this trial seems to be developed with careful thought and the authors provide a detailed report.

Reply: We want to thank the reviewer for thorough reading of this manuscript and the insightful comments. We appreciate the kind words about the trial.

Major comments:

-Can the authors justify why they are examining a composite outcome of a host of problems? While all of the outcomes are important, it seems odd to include functional status (SF-36) with harder endpoints such as surgical complications and death. A higher weight for death vs. physical function in the composite outcome is interesting... why wouldn't physical function be more highly weighted? The patient themselves may be more devastated if they end up in worse health after their surgical intervention.

Reply: Indeed, we examine different outcomes in a composite endpoint. As described in the discussion 'the current program is multidisciplinary targeting different aspects of surgical outcomes in patients undergoing cardiac surgery. Because different aspects are targeted, a composite weighted score will better reflect the treatment benefits than a single outcome.' (p.20, Line 357-360). The reviewer is correct that physical function becomes a more important issue for patients. However, mortality and morbidity are still important from a patient perspective. Patients weighted the risk of death as more important (relative weight 0.23) compared to stroke or myocardial infarction (resp .18

and .14), which both can impair physical function strongly[1]. Therefore mortality was weighted with the highest score of three. Since there are two scales for physical function, a patient with worse physical function after surgery will consequently score two, which we think is a sufficiently weighted score. As described in the discussion we think the combination of physical function with hard endpoints such as surgical complications and death will better reflect a comprehensive representation of the recovery of the patients than choosing one of these outcomes alone (P. 20, Line 357-363). To ensure transparency and to enable different interpretations of the main outcomes, as implied by the reviewer, we will also report the outcomes of the individual components and unweighted. Finally, we think that physical function is not underexposed in this protocol, because various aspects (i.e. muscle strength, cardiorespiratory fitness and independence in activities of daily living) of physical function are measured over multiple time points.

1. Tong BC, Huber JC, Ascheim DD, Puskas JD, Ferguson Jr B, Blackstone EH, et al. Weighting Composite Endpoints in Clinical Trials: Evidence for the Heart Team. 2013;94(6):1908–13.

-This study protocol is similar to one previously published in BMJ Open (PMID: 25753362) and to a small study examining prehab (PMID: 24459173) which were not cited in the author's paper. Can the authors please comment on how their prehab intervention will build on these studies and the ones cited in their document (ref 23, 36)?

Reply: The study protocol published in the BMJ Open (PMID: 25753362) is indeed an interesting study with a well-designed protocol. We are looking forward to read the results which are, to our knowledge, not published yet. 'One preoperative CR study and' has been added to the sentence 'Previous preoperative CR studies were primarily focused on short-term effects; only one preoperative CR study and a few post-operative CR studies have determined long-term effects.(23,35,36,72)' (P.20,21, Line 379-381)

The small study examining prehab (PMID: 24459173) showed an improvement in physical fitness in patients following a Prehab intervention. 'Physical fitness' has been added to the sentence 'In addition to post-operative CR, small trials suggested that preoperative CR is effective in reducing post-operative pulmonary complications, duration of hospital stay, improving HRQoL, physical fitness and increasing the compliance to post-operative CR.(23,36,37)' (P5., Line 98-101).

The studies examining prehab mentioned by the reviewer, are mainly focused on preoperative rehabilitation including inspiratory muscle training (IMT) or exercise and education, whereas the Heart-ROCQ study evaluates a program in which the pre-and postoperative rehabilitation is combined. In addition, the Heart-ROCQ program combines the investigated components (e.g. IMT, aerobic cycling and, education) in previous studies with resistant training, psychological and dietary guidance. Moreover, coaching to stop smoking and to return to work, are available for patients who respectively smoke or are employed.

Thus, the Heart-ROCQ program is more extensive with the different types of sessions, but also has a higher frequency (3 times per week vs. 2 times per week). In this way, the program might be more effective in a shorter amount of time compared to the programs in previous studies (min. 3 weeks vs. 8-16 weeks).

In addition to previous studies, we perform an economic evaluation to learn more about the cost-effectiveness of pre- and postoperative rehabilitation. Moreover, we examine mortality up to five years after surgery.

Minor comments:

-It may be helpful to revise the document for English language to improve some clarity within the document.

Reply: Thank you for this observation. We had the manuscript read by someone who is native in the (UK) English language, who revised the manuscript for English language (textual changes are highlighted in yellow)

-Please clarify in the title that this is a study protocol.

Reply: As suggested by the reviewer, 'study protocol for' is added to the title to clarify that this manuscript is a study protocol.

-The use of CS as an acronym throughout the paper does not help with reading the document. Can the authors please spell out cardiac surgery instead of using CS?

Reply: As suggested by the reviewer, we spelled out cardiac surgery throughout the manuscript.

-Where it says, "in a pilot study (to be submitted) the patient satisfaction and the feasibility (in terms of accessibility, compliance, training load and safety) of the Heart-ROCQ program have been evaluated. These results were taken into account when further developing the Heart-ROCQ program and the protocol of this study." Can the authors please provide a summary (e.g., in a table, a box, or in text) how their study was tailed based on the pilot study in the protocol here? It would be useful for other readers to know this specifically in this document.

Reply: We agree the observation that it would be useful to other readers. Therefore we added to the PPI statement how the results of the pilot study were taken into account in the development of the study protocol ('Patients were very satisfied with the program ---- in the further development of the Heart-ROCQ program and the protocol of this study.', P.18,19, Line 338-349). As advised, the PPI statement is placed at the end of the method section.

-The authors should indicate how their primary endpoint will be analyzed throughout the trial.

Reply: The components of the primary endpoints are adjudicated by the independent endpoint committee, blinded for group allocation (P. 10, Line 233, 234). Throughout the trial, the primary endpoint will only be analyzed when the interim analysis is conducted and at the end of the study. As described in the methods: 'An interim analysis will be conducted when 40% of the included patients have had the measurements one year after surgery.' (P. 17, Line 306-309).

VERSION 2 – REVIEW

REVIEWER	Scott Kehler Dalhousie University, Canada
REVIEW RETURNED	23-Jul-2019
GENERAL COMMENTS	I would like to thank the authors for addressing the editor and reviewer comments. One last point: it is still unclear to me what specific statistical test(s) will be used to analyze their outcomes. For example, will a two-way repeated measures ANOVA be used? A linear mixed model? The quoted text that the authors provide in their response may be too non-specific for readers

VERSION 2 – AUTHOR RESPONSE

Reviewer(s)' Comments to Author:

Comments from Reviewer: 1

Reviewer Name: Scott Kehler

Institution and Country: Dalhousie University, Canada

Please state any competing interests or state 'None declared': None declared

I would like to thank the authors for addressing the editor and reviewer comments. One last point: it is still unclear to me what specific statistical test(s) will be used to analyze their outcomes. For example, will a two-way repeated measures ANOVA be used? A linear mixed model? The quoted text that the authors provide in their response may be too non-specific for readers

Reply: Thank you for your attentiveness. The total score of the primary endpoint will be handled as a continuous variable. A linear mixed model will be used with treatment group (i.e. Heart-ROCQ group and Standard Care group) as between factor and the measurements as within factor. We added the following sentences to the heading '*Primary and sensitivity analyses*':

'The total score of the primary endpoint will be handled as a continuous variable. All continuous variables will be analysed using linear mixed models to determine 'time x group' differences. Significant interactions will be further explored using the Bonferroni post-hoc test to determine differences between each time point. Non-parametric tests will be used if the assumptions of normal distribution are violated.' (P.15, Line 307-311)

VERSION 3 – REVIEW

REVIEWER	Scott Kehler Dalhousie University, Canada
REVIEW RETURNED	22-Aug-2019
GENERAL COMMENTS	The authors have sufficiently addressed my comment